# Influence of Beam Angle on Normal Tissue Complication Probability of Knowledge-Based Head and Neck Cancer Proton Planning

**DOI:** 10.3390/cancers14122849

**Published:** 2022-06-09

**Authors:** Roni Hytönen, Reynald Vanderstraeten, Max Dahele, Wilko F. A. R. Verbakel

**Affiliations:** 1Varian Medical Systems Finland, 00270 Helsinki, Finland; 2Varian Medical Systems Belgium, 1831 Diegem, Belgium; reynald.vanderstraeten@varian.com; 3Department of Radiation Oncology, Amsterdam UMC Location Vrije Universiteit Amsterdam, 1081 HV Amsterdam, The Netherlands; m.dahele@amsterdamumc.nl (M.D.); w.verbakel@amsterdamumc.nl (W.F.A.R.V.); 4Cancer Center Amsterdam, 1081 HV Amsterdam, The Netherlands

**Keywords:** knowledge-based planning, intensity-modulated proton therapy, normal tissue complication probability, automated optimization

## Abstract

**Simple Summary:**

Knowledge-based treatment planning (KBP) solutions can be used to assist in the planning process by automatically generating patient-specific optimization objectives. A KBP model is typically derived from treatment plans with a similar planning methodology, for example beam angles. An end-user might deviate from this methodology for a variety of reasons. The effect of such deviations on KBP plan quality has not been widely explored. We therefore studied this using a human-interaction free proton planning solution to create comparative plans with the default angles used when building the model, and altered beam angle arrangements. Because normal tissue complication probability (NTCP) can be used to select patients for proton therapy, this was used as the primary outcome metric for plan quality. The results show that the beam angle and number of beams only had a small effect on the plan NTCP. This suggests that the model is robust to the various beam arrangements within the range described in this analysis, although a method that automatically further adapts the KBP planning objectives further decreased the NTCP by 1–3%.

**Abstract:**

Knowledge-based planning solutions have brought significant improvements in treatment planning. However, the performance of a proton-specific knowledge-based planning model in creating knowledge-based plans (KBPs) with beam angles differing from those used to train the model remains unexplored. We used a previously validated RapidPlanPT model and scripting to create nine KBPs, one with default and eight with altered beam angles, for 10 recent oropharynx cancer patients. The altered-angle plans were compared against the default-angle ones in terms of grade 2 dysphagia and xerostomia normal tissue complication probability (NTCP), mean doses of several organs at risk, and dose homogeneity index (HI). As KBP could be suboptimal, a proof of principle automatic iterative optimizer (AIO) was added with the aim of reducing the plan NTCP. There were no statistically significant differences in NTCP or HI between default- and altered-angle KBPs, and the altered-angle plans showed a <1% reduction in NTCP. AIO was able to reduce the sum of grade 2 NTCPs in 66/90 cases with mean a reduction of 3.5 ± 1.8%. While the altered-angle plans saw greater benefit from AIO, both default- and altered-angle plans could be improved, indicating that the KBP model alone was not completely optimal to achieve the lowest NTCP. Overall, the data showed that the model was robust to the various beam arrangements within the range described in this analysis.

## 1. Introduction

The radiotherapy treatment planning process is laborious, subjective, and prone to both inter- and intra-institutional variations [1,2]. Knowledge-based planning tools have been shown to improve the consistency and quality of both photon and proton treatment plans, while reducing the planning time [3,4,5,6,7]. With these tools, the achievable dose distribution for a prospective patient can be estimated based on a predictive model trained with prior treatment plans [3,8]. One of the commercially available proton knowledge-based planning systems is RapidPlan for protons (RapidPlanPT, Varian Medical Systems, Palo Alto, CA, USA), which uses parametrizations of the plan geometry and associated dosimetry to produce dose–volume histogram (DVH) predictions [4]. The predictions can subsequently be used to dictate the placing of the optimization objectives, partially automating the planning process, and producing a knowledge-based plan (KBP). This approach has been shown to be capable of producing good quality plans for complicated cases, such as head and neck cancer (HNC), when the model is properly trained [9,10,11]. However, the RapidPlanPT model is typically created using plans from a single institution. These plans have usually been made using a particular planning technique, with certain beam arrangements. For individual patients, RapidPlanPT users may want to use different beam arrangements for a variety of reasons, e.g., their own institutional preference, patient geometry, or avoidance regions.

It has already been demonstrated that if the geometric data differ significantly from the training data, plans may be suboptimal [8], and if the model has been trained on plans generated according to an older, potentially inferior optimization protocol, the KBP is likely to require additional optimization [12]. With regards to what is known about the influence of beam arrangements, Xu et al. recently demonstrated that a RapidPlanPT model trained with HNC plans spanning a wide range of beam arrangements can produce plans of comparable or better quality than expert plans, when applied to customized beam arrangements [10]; and Delaney et al. have previously demonstrated the efficacy of RapidPlanPT in cases where the model was applied to prospective cases with only limited (gantry angles at 35–55°, 180°, and 305–330°) variation in beam angles from the training set [9]. The influence of beam arrangement on KBP model performance remains to be investigated. Therefore, in this study, we analyze the performance of a model trained on plans generated using a standard beam arrangement, when applied to plans with substantially altered beam arrangements (variation in beam angle and beam number). Because normal tissue complication probability (NTCP) is being used to select patients for proton therapy, this was used as the primary outcome metric for plan quality [13,14].

If RapidPlanPT produces suboptimal KBPs when applied to plans with non-standard beam arrangement, it should be possible to optimize these plans further than their default-arrangement counterparts. For this purpose, we introduce a proof of principle automatic iterative optimizer (AIO), implemented using the Eclipse Scripting Application Programming Interface (ESAPI, Varian Medical Systems, Palo Alto, CA, USA) [15,16], tasked to maximally reduce the KBP normal tissue complication probability (NTCP) while maintaining acceptable planning target volume (PTV) coverage. While there have been numerous publications on both commercial and in-house automatic optimizers, we found only a few demonstrating their implementation using generally available tools such as RapidPlan (PT) and ESAPI, especially for HNC proton treatment planning [3,8,17,18,19,20,21,22].

In summary, in this study, we use a previously introduced automated knowledge-based planning solution and a previously validated RapidPlanPT model to generate KBPs with both default and altered beam angle arrangements without human interaction [12]. We demonstrate the AIO process and evaluate the default and altered beam arrangement KBPs both before and after AIO in terms of NTCP and the PTV homogeneity index (HI).

## 2. Materials and Methods

### 2.1. Patient Cohort and KBP Creation

Ten recent HNC patients with locally advanced oropharynx cancer, previously treated with 2-arc VMAT, were arbitrarily selected for the study. All patients had given signed consent that their data could be used within the department’s research program, and the institutional medical ethics committee exempted this work from requiring their official approval.

All plans in this study were automatically generated using the previously described automated knowledge-based planning solution (Figure 1a) [12]. In brief, the solution uses the Eclipse scripting application programming interface (ESAPI) to generate proton or photon KBPs based on patient CT and delineation [15,16]. The process is controlled with plan templates, indicating the desired prescription, beam setup, etc., and it leverages RapidPlanPT to generate knowledge-based optimization objectives. In this study, the plan templates were configured to produce a default-arrangement plan (coplanar beams at 45, 180 and 315 degrees) and eight altered-angle plans with either a 45- or 315-degree beam altered by ±20 or ±40 degrees (Figure 1b). Additionally, for three patients, we created 4- and 5-field plans to study the RapidPlanPT and AIO performance with a larger number of beams than in the training set. In the 4-field plans, the additional field was added at ± 90 degrees ipsilaterally to the PTV to minimize the path length in body. Similarly, in the 5-field plans, the additional fields were added at ±135 degrees to maximize the PTV coverage from different directions. The plans used a simultaneously integrated boost technique, delivering 70/54.25 Gy to the boost/elective planning target volumes (PTV_B_/PTV_E_) in 35 fractions. Additionally, a 5 mm wide ring (PTV_O_) around PTV_B_ was subtracted from PTV_E_ to facilitate a steep dose fall-off.

The KBPs with RapidPlanPT-generated line objectives were non-robustly optimized using Varian Eclipse nonlinear universal proton optimizer (NUPO) 16.0.2 with multifield optimization using proximal/distal/lateral target margins of 2/3/5 mm. The OAR dose objectives were as follows: maximum of 50 and 54 Gy to spinal cord and brain stem expanded with 3 mm margin, respectively, but preferably lower; as low as possible mean dose to both parotid and submandibular glands, oral cavity, and individual swallowing muscles [23].

The dose calculation was performed using proton convolution superposition algorithm 16.0.2, and the plans were normalized to PTV_B_ V_95%_ = 98%. The RapidPlanPT model had been trained on 50 manually planned 3-field IMPT plans based on patients not included in this study, and it has previously been shown to produce high quality KBPs [11].

### 2.2. Automatic Iterative Optimizer

We designed a proof of principle automatic iterative optimizer (AIO) solution for further optimization of the KBPs in an iterative manner. When performed manually, the treatment planner would adjust the optimization objectives to find the lowest achievable OAR doses and the trade-off between OAR sparing and PTV dose homogeneity [18,24]. AIO was implemented using the ESAPI scripting interface, and it aims to simulate this planning process in an automated manner.

In order to prevent underdosing of parts of the target overlapping with OARs in the AIO-produced plans, a set of helper structures were introduced by splitting the NTCP-related OARs into portions within and without PTV_B_ plus margin (OAR_in_ and OAR_out_, respectively). These structures are used in the AIO process to drive the dose down in OAR_out_ while avoiding both underdosing and hot spots within OAR_in_.

As a starting point, AIO uses an optimized treatment plan, a KBP in this case, checking that the plan has a sufficient margin in PTV_B_ maximum dose (D_Max_) to facilitate the optimization process, as reduced PTV homogeneity can lead to significantly lower OAR doses [25]. If the plan passes the initial check, additional mean dose objectives for OAR_out_ and maximum dose objectives for OAR_in_ are added. The mean dose objectives of the OAR_out_ are set to the respective mean dose minus a reduction dose (*x*) with priority comparable to the pre-existing OAR objective priorities. Likewise, the OAR_in_ objectives are set to PTV_B_ target dose with priorities comparable to the PTV_B_ objective priorities. Underdosing in the PTV_B_ is avoided by giving its minimum dose objectives higher priority than the OAR_in_ objectives. The initial value of reduction dose *x* is set to 0/2.5/5 Gy, depending on the initial dose homogeneity so that the more homogeneous plans are optimized more aggressively.

Once the new optimization objectives are added, the plan is optimized, followed by dose calculation and normalization using the process described for KBP creation. If the resulting plan has not reached the homogeneity threshold (i.e., PTV_B_ D_Max_ < 109%) and there is little reduction in OAR D_Mean_, the reduction dose is increased for all OAR_out_. If the objective reduction was too aggressive, i.e., PTV_B_ D_Max_ > 110%, the reduction factor is reduced. Finally, the mean dose objectives are updated to D_Mean_–*x* according to the latest optimizer dose, and the process is iterated until the PTV_B_ homogeneity threshold is reached. The resulting treatment plans (AIO-KBPs) are expected to have better NTCP values with slightly degraded dose homogeneity.

### 2.3. Plan Evaluation

The altered-angle KBPs and AIO-KBPs were compared to the default-angle ones by computing the PTV_B_ homogeneity index (HI), conformity index (CI), and plan NTCP. The HI and CI were defined respectively as
(1)$$\mathrm{HI}=\frac{D_{2\%}-D_{98\%}}{D_{p}}\times 100$$
(2)$$\mathrm{CI}=\frac{V_{95\%}}{V_{\mathrm{PTV}}},$$
where D_x%_ is the dose to x% of the PTV_B_, D_p_ is the prescription dose to PTV_B_ (70 Gy), V_x%_ is the isodose volume receiving at least x% of dose, and V_PTV_ is the volume of PTV_B_. The NTCPs (grade 2 and 3 dysphagia and xerostomia at 6 months after the treatment) were evaluated according to models adopted by the Dutch radiation oncology society [26]. Model details and the parameters used are presented in Appendix A. For evaluation purposes, we represent the sum of grade 2 NTCP endpoints as NTCP_Σ_.

The HI and NTCP were also used to compare AIO-KBPs to the respective KBPs, and to study whether the altered- and default-angle plans behaved differently in the optimization. The statistical significance of the results is evaluated using two-sided Wilcoxon signed rank test at *p* < 0.05.

## 3. Results

### 3.1. Default- and Altered-Angle KBPs

The KBPs with altered beam angles produced by the automated knowledge-based planning solution, before AIO was applied, had on average slightly worse NTCP metrics and a slightly better homogeneity index (Figure 2 and Figure 3). The patient-specific grade 2 dysphagia and xerostomia were lowest for the default beam angle arrangement in 42 and 61 out of 80 cases (10 patients, each with 8 altered beam angles), respectively. The respective mean differences were 0.3 ± 1.0 and 0.3 ± 0.4% in favor of default-angle KBPs, where only the latter was statistically significant. Both grade 2 dysphagia and xerostomia were reduced the most for patient seven, with respective reductions of 1.2 and 0.3% in two separate KBPs with beams at 25° and 85°. There were no statistically significant differences in NTCP_Σ_, HI, or CI between default- and altered-angle KBPs, and the altered-angle plans showed at most <1% reduction in NTCP_Σ_. However, there were 13 altered-angle plans with HI reduction >1.

For all OAR, the differences in default- and altered-angle KBP mean doses were mainly between ±2 Gy, with individual plans seeing differences up to ±6 Gy in the oral cavity, superior pharyngeal constrictor muscle, and spinal cord. A graph of OAR-wise mean dose results is presented in Appendix A. There were no significant differences between 20- and 40-degree beam angle alterations, and the best arrangement varied from patient to patient. Exemplary NTCP and OAR-specific mean dose results over all plans of two patients are presented in Figure 4, demonstrating cases where alteration of the beam angle has little to no effect, and where it has some effect. The respective patient delineations are presented in Appendix A.

### 3.2. KBPs and AIO Plans

AIO managed to reduce NTCP_Σ_ in 66 out of 90 KBPs, and improved NTCP_Σ_ by an average of 3.5 ± 1.8% (Figure 5). Generally, adding AIO lead to a larger reduction in NTCP than changing the beam angle, although in some cases, altered-angle KBP + AIO led to lower ∆NTCP_Σ_ values (Figure 3). Dysphagia benefitted from AIO slightly more than xerostomia, especially with regard to grade 3 endpoints. As a trade-off, the average HI of the AIO plans increased by 0.2 ± 0.7, the average CI increased from 1.09 ± 0.03 to 1.18 ± 0.06, and the average PTV_B_ D_Max_ increased from 74.8 ± 0.9 Gy to 76.9 ± 0.3 Gy. All differences were significant at *p* < 0.01. On average, AIO improved the mean doses of NTCP-related OAR in about 75% of the plans (Figure 6a,b). In most AIO plans, the OAR mean doses relevant for NTCP were reduced by less than 5 Gy (Figure 4). Individual plans saw mean dose reductions of up to 15 Gy in the inferior pharyngeal constrictor muscle (PCM Inf) and the submandibular gland contralateral to PTV_B_ (Subm C).

None of the OAR were systematically reduced in any of the AIO plans, but were instead increased by <1 Gy in about 25% of cases. None of the OAR (NTCP-related or otherwise) saw mean dose increases over 5 Gy, apart from the oral cavity, the mean dose for which was increased by over 5 Gy in two cases (Figure 6c,d).

On average, AIO converged in about six iterations, which corresponds to about 20 min of processing time.

### 3.3. Default- and Altered-Angle AIO Plans

The NTCP values in AIO plans were more evenly distributed, with 45 and 40 out of 80 grade 2 dysphagia and xerostomia endpoints, respectively, being lower for the default-angle plans (as compared to 42 and 61 in KBPs, Figure 2). The maximum NTCP_Σ_ reduction achievable with altered beam angles also increased from <1 to almost 4% (Figure 3). The AIO process was slightly more effective in reducing the NTCP_Σ_ for altered-angle plans, with a mean reduction of 3.6 ± 1.7%, as opposed to 2.9 ± 2.0% for default-angle plans (*p* < 0.05). The largest variation in dysphagia over the beam angles was for patients six and seven, where the size of PTV_B_ and its location in relation to the pharyngeal constrictor muscles and oral cavity increases the effect of beam angles on the NTCP. Figure 5 shows how the mean dose to different OAR varied with beam angles for patients six and five, where patient five exemplifies a case in which beam angles had little to no effect.

Additionally, the HI was increased less in altered-angle plans, with a mean increase of 0.2 ± 0.7, as opposed to 0.6 ± 1.0 for default-angle plans (*p* < 0.05), indicating that the optimizer was forced to make more aggressive trade-offs between PTV homogeneity and OAR dose in default-angle plans. As with pre-AIO KBPs, the best beam angle arrangement varied from patient to patient, but the differences were slightly more noticeable in the AIO plans (Figure 5). There were no statistically significant differences in CI between the default- and altered-angle plans.

### 3.4. Increased Number of Beams

When applied to prospective beam arrangements with increased number of fields, the performance of both RapidPlanPT and AIO varied. Increasing the number of beams to four or five improved NTCP_Σ_ and HI slightly (by <1.5% and <1, respectively) but had little to no effect on CI, with all plans having a CI between 1.03 and 1.06. The AIO process successfully reduced NTCP_Σ_ in 5/6 KBPs, with a maximum reduction of 3.9%. The effect of AIO processing on HI was inconclusive; in 2/3 patients, HI was increased by <1, but for one patient it was reduced by a similar amount. The CI was increased on average by 0.05 for all plans. A graph of the results is presented in Appendix A

## 4. Discussion

This study evaluated the performance of RapidPlanPT when creating KBPs with beam angle arrangements differing from those in the training set. A previously demonstrated RapidPlanPT model and automated KBP-creation pipeline, as well as a novel proof-of-principle automated iterative optimizer (AIO), were employed to create a total of 90 KBPs for 10 HNC patients without human interference. Our results demonstrate that changing the beam angles could occasionally lead to <1% lower NTCP_Σ_, but on average, the standard beam angles were the best choice. Adding AIO further reduced the NTCP_Σ_ on average by about 3% for all beam angles, depending on the initial quality of the KBP. Increasing the number of fields did not substantially lower the NTCP_Σ_.

To place this study in context, RapidPlanPT has been previously studied by Delaney et al. and Xu et al., demonstrating its clinical feasibility with models trained on plans with standardized (former) and varying (latter) beam angle arrangements [9,10]. Our study sits, in a sense, between these two approaches, and builds on top of our automated knowledge-based planning framework, further expanding it [12]. Multiple authors, including Tol et al. and Breedveld et al., have previously demonstrated different solutions to automate the treatment plan optimization process [8,16,17,18]. Our approach is most akin to what was suggested by Tol et al., with the main difference that we interact with the optimization engine through the scripting interface instead of the graphical user interface layer, which should make it easier to implement.

For 13/80 altered-angle AIO plans, there were NTCP_Σ_ reductions of 2–4%. This may be clinically relevant when set against the threshold of a 15+% reduction in NTCP_Σ_ (for grade 2+ toxicity) that can lead to patients being selected for proton therapy in the Netherlands [13]. Such data argue for more aggressive approaches to maximizing treatment plan quality (e.g., integration of AIO into the KBP process). These larger reductions were associated with patients that had smaller PTV_B_ volumes, and thus less overlap between PTV and NTCP-relevant OARs (e.g., Patient 6). This would indicate that, for clinical practice, where manually choosing and testing even a moderate number of different beam angles is time consuming and laborious, there is little benefit to be gained for NTCP from fine-tuning the patient-specific beam angles. Instead, using a template arrangement, and performing crude alterations to it based on the relative OAR location and potential avoidance regions, such as the dental amalgam, would be more efficient. Automated beam angle optimization, while expected to confer a small benefit, nonetheless merits consideration as a part of a comprehensive approach to automation and plan-quality improvement [27]. In addition, the scripting would also allow for an automatic investigation of whether beam angle changes of less than 20 degrees could lead to NTCP improvements.

We also studied the effect of increasing the number of beams to four or five by generating these KBPs for three patients. Earlier studies have found a larger number of fields to have either minor or no effect on OAR sparing, PTV coverage, or plan robustness [28,29,30]. Our results are consistent with these findings, showing a <1% reduction in NTCP_Σ_ per added field, even after the AIO process. The effect on homogeneity was inconclusive, with KBPs showing little to no benefit, and AIO plans showing moderate improvements for two patients, and a worse HI for the third.

The major limitations of this study were the lack of robust optimization and the associated beam path considerations, as well as the plans being optimized for lowest OAR dose instead of for the highest homogeneity or conformity and lowest NTCP. Optimizing the plan robustly could lead to more significant differences between beam angle arrangements, especially in cases where the beam path is through highly inhomogeneous tissue, or there are critical OAR close to the target [29]. Similarly, if the target shape and potential avoidance regions, e.g., critical organs and metal implants, had been considered, and plans had been optimized specifically for lowest NTCP and high homogeneity, the results could have differed.

Our proof-of-principle AIO process was able to improve the NTCP_Σ_ in 66 out of 90 KBPs, reducing it by 2.9 and 3.6% for default- and altered-angle KBPs, respectively. This highlights that there is frequently room for improvement in plans and presumably reflects the fact that the plans used to generate the KBP model were themselves not maximally optimized. Such data highlight a wider role for approaches such as AIO, which not only emphasize automation (which is no guarantee of quality) but also focus on the generation of high-quality patient-specific plans. While the use of KBP is generally advised against in the case of plans that are considered outliers (e.g., PTV size/location, OAR size/location) because this would create sub-optimal optimization objectives, adding AIO could probably solve this problem. The plans that could not be improved either had high initial PTV_B_ D_Max_, leading to them being directly discarded from AIO (13 plans), or were ones in which the first iteration failed to find a solution that would reduce NTCP_Σ_ without violating the 110% PTV D_Max_ threshold (11 plans). In future, a separate AIO routine to increase PTV_B_ homogeneity at the cost of OAR dose could be realized to address these plans. On average, the process took about six iterations, or 20 min, to converge. The maximum number of iterations was capped at 10, as it was found that after this point there was very little improvement, especially when used together with a more aggressive initial reduction dose for plans with low initial PTV_B_ D_Max_. A major bottleneck in the process was, however, the need for dose calculation following each optimization round to check whether the end conditions had been met. This was due to the optimizer-reported maximum dose differing from the calculated and normalized dose by more than our 1% target margin. If the AIO process could be made less sensitive to this difference, and the dose calculation steps could be omitted from the iteration process, the processing time would fall by about 25%.

In addition to speed, AIO would benefit also from other improvements to make it more clinically relevant: If the optimization objectives could be adjusted during the optimization engine processing, as is possible via the graphical user interface of NUPO, a lot of time could be saved on not having to instantiate it between each iteration (i.e., transferring the plan data from the client to the optimizer only once). Furthermore, a more holistic solution for constraints and end conditions would be required to ensure that an acceptable plan quality is maintained. One possibility, and an interesting topic for future research, would be a clinical goals-based “wish-list” of dosimetric objectives that AIO could seek to fulfill during processing [17]. While we applied AIO exclusively to proton planning in this work, it should be applicable to photon optimization without major modifications: this merits further evaluation.

## 5. Conclusions

A KBP model to create head and neck cancer proton plans has been shown to be robust to beam arrangements that differ substantially from those used in the plans on which the model was based. Further automated optimization of KBP plans with default and altered beam angles showed that many could be improved to some extent or another. This suggests that the plans used to generate the model were not fully optimized for reduction of specific NTCP and HI parameters. Furthermore, plans consisting of more beams could be generated using KBP + AIO, but increasing the number of beams only led to clinically insignificant reductions in the NTCP.

## Figures and Tables

**Figure 1 cancers-14-02849-f001:**
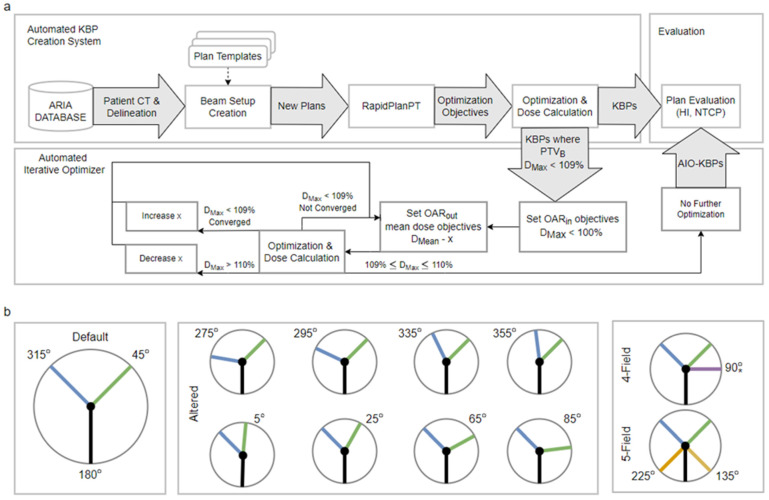
(**a**) KBP-creation and AIO-processing workflow. (**b**) Illustration of beam angle arrangements. The angles not separately indicated in the middle- or right-pane diagrams are the same as in default arrangement. * In the 4-field arrangement the additional field is added ipsilaterally of the PTV, i.e., either to 90 or 270 degrees. KBP = knowledge-based plan; AIO = automatic iterative optimizer; HI = homogeneity index; NTCP = normal tissue complication probability; OAR = organ at risk.

**Figure 2 cancers-14-02849-f002:**
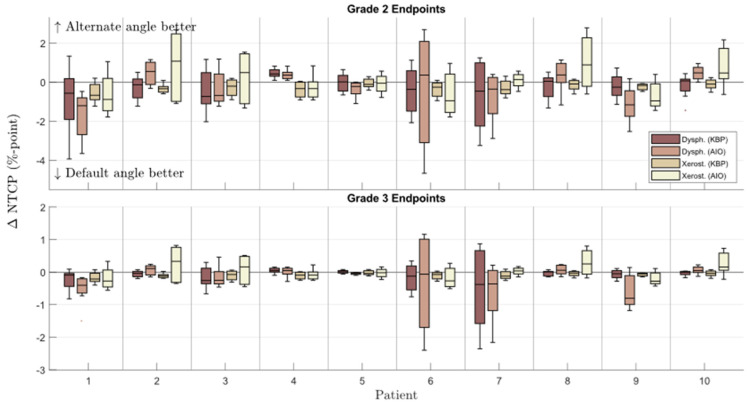
Box-whisker plot of patient-wise NTCP difference between KBPs/AIO plans with default and alternate beam angles (default minus altered). Dark line in the middle of the box indicates the median; the top/bottom of the box indicates 75th/25th percentile; and the whiskers indicate the range of the data. Dysph. = dysphagia; Xerost. = xerostomia; KBP = knowledge-based plan; AIO = automatic iterative optimizer-generated plan; ∆NTCP = normal tissue complication probability difference.

**Figure 3 cancers-14-02849-f003:**
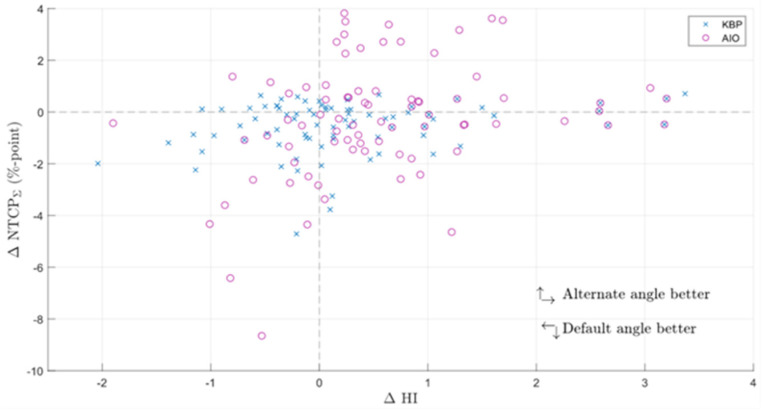
Difference in the sum of grade 2 NTCP endpoints and HI between KBPs with default and alternate beam angles (default minus altered). ∆NTCPΣ = difference in the sum of grade 2 normal tissue complication probabilities; KBP = knowledge-based plan; AIO = automatic iterative optimizer-generated plan.

**Figure 4 cancers-14-02849-f004:**
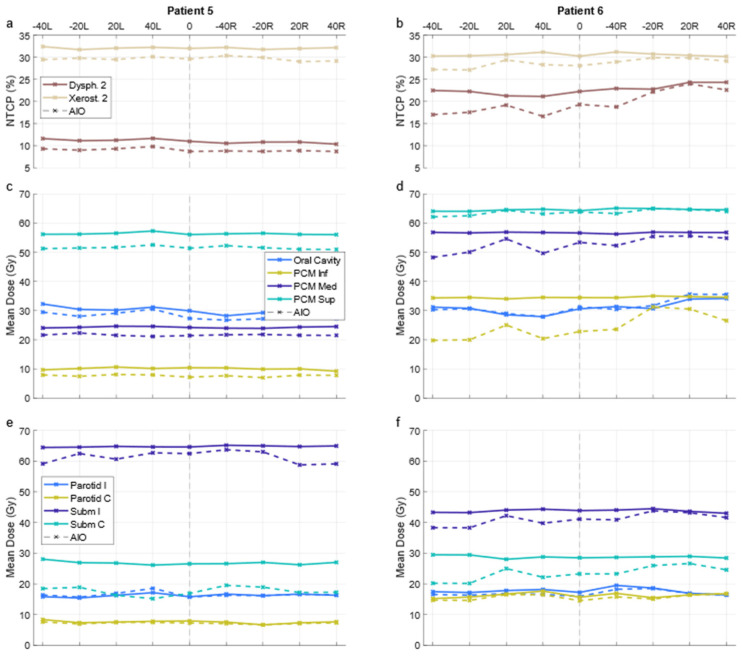
NTCPs (**a**,**b**) and OAR mean doses (**c**–**f**) over all plans for two patients. Solid and dashed lines indicate KBP and AIO plans, respectively. X-axis indicates the angular deflection of one of the fields; “0” corresponds to the default arrangement and “−40R” to the plan where the right field (in patient coordinates) is deflected by −40 degrees. PCM Inf/Med/Sup = inferior/middle/superior pharyngeal constrictor muscle; Parotid I/C = ipsi-/contralateral parotid gland; Subm I/C = ipsi-/contralateral submandibular gland; KBP = knowledge-based plan; AIO = automatic iterative optimizer-generated plan.

**Figure 5 cancers-14-02849-f005:**
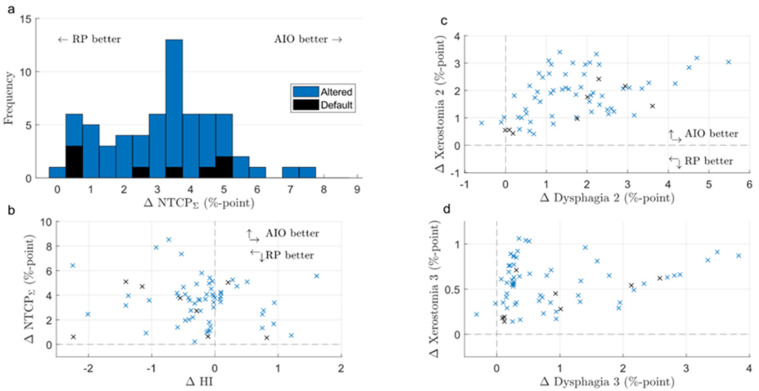
Difference in NTCP_Σ_ (**a**), NTCP_Σ_ and HI (**b**), and individual NTCP grade 2 and 3 endpoints (**c**,**d**) between RP- and AIO-produced KBPs (RP minus AIO). The upper right quadrant of scatter plots indicates that both endpoints are superior in AIO plans. RP = Rapid Plan-generated plan; AIO = automatic iterative optimizer-generated plan; ∆NTCPΣ = difference in the sum of grade 2 normal tissue complication probabilities.

**Figure 6 cancers-14-02849-f006:**
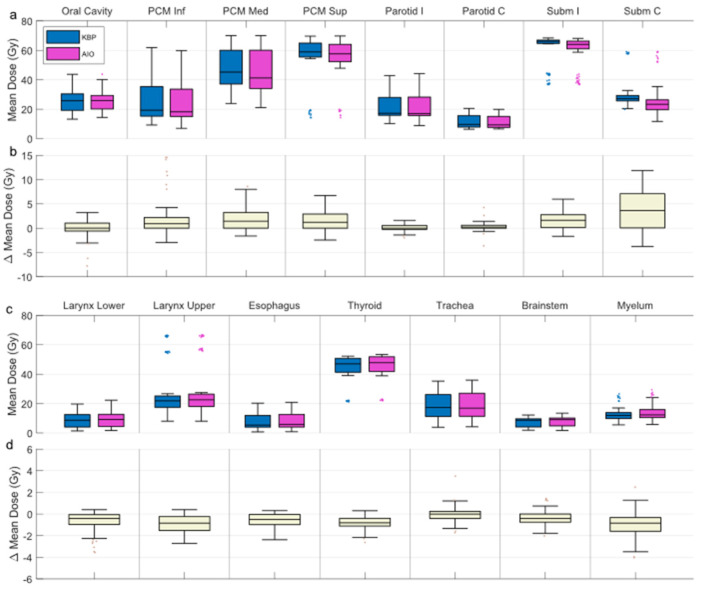
Box-whisker plot of OAR mean doses for all KBPs and AIO plans (**a**,**c**), and the plan-wise difference between the two (**b**,**d**; KBP minus AIO). Dark line in the middle of the box indicates the median; the top/bottom of the box indicates 75th/25th percentile; and the whiskers indicate the range of the data. PCM Inf/Med/Sup = inferior/middle/superior pharyngeal constrictor muscle; Parotid I/C = ipsi-/contralateral parotid gland; Subm I/C = ipsi-/contralateral submandibular gland; KBP = knowledge-based plan; AIO, automatic iterative optimizer-generated plan.

## Data Availability

The data presented in this study are not available at this time.

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
