# Peer review of "Influence of Beam Angle on Normal Tissue Complication Probability of Knowledge-Based Head and Neck Cancer Proton Planning"

_cancers, 2022, doi:10.3390/cancers14122849_

Round 1

Reviewer 1 Report

General comments:

In this study, the performance of a knowledge-based planning (KBP) model together with an automatic iterative optimizer (AIO) of a commercial proton therapy TPS in generating automatically treatment plans with both default and altered beam angle arrangements for 10 oropharynx cancer patients previously treated with VMAT was studied. The quality of the plans generated with 4 planning conditions, i.e. with default and altered beam arrangements and before and after application of AIO, were evaluated based on NTCP and PTV dose homogeneity metrics.  

A fair comparison between the treatment plans generated with the 4 planning conditions would depend on whether the individual plans were optimally optimized. The quality of an optimal treatment plan could be affected by the shape, volume, and location of the target volume. Beam angle could become important when dealing with PTV of highly irregular shape and close to or overlapping with critical OARs. The question is would the finding of the study be different if beam angles were taken into account in dose optimization or if the patients were categorized, for instance, according to complexity of target geometry.

Range uncertainties was also important in PT planning, especially when complex anatomy and highly inhomogeneous tissue structures were involved in the beam path.  This would have an impact on optimal beam angle selection and indicate the need for beam angle optimization. This issue was not addressed in the manuscript.

In this study, NTCP and HI were the key quality metrics for assessment of quality of treatment plans. Apart from dose inhomogeneity, target dose conformity of the target was also an important metric of a quality treatment plan. It was not indicated in the manuscript if this important metric had been included in plan optimization and plan assessment.

Specific comments:

Line 160: Supplementary Material X was not provided. It appears that this was a typo.

Line 162: HI and NTCP were used to compare AIO-KBPs to the respective KBPs. Dose conformity of PTVB should also be used in the comparison.

Figure 1(b): It is not mentioned why the beam angles indicated were selected for the 4th and 5th beams.

Line 207: (Subm C) should be defined.

Author Response

We would like to thank the reviewer for their comments, which we are sure will strengthen the article.

We have made the following changes based on the specific comments:

  • We corrected the typo in ‘Supplementary Material X’ to now read ‘Supplementary Material B’.
  • We have explained our reasoning for selecting the specific beam angle arrangements for 4- and 5-field plans; to minimize the beam path in body and to maximize the coverage of the fields, respectively.
  • We have defined Subm C more clearly to be the submandibular gland contralateral to the PTVB.
  • We have included dose conformity index (CI) to the list of properties we evaluate for the plans. The CI is now defined in section 2.3, and the results are given in sections 3.3 and 3.4.

We have also now addressed the role of robust optimization and range uncertainties in the discussion.

Reviewer 2 Report

1. This is a cross-country study with novelty. KBP has previously been deemed a useful tool in the planning process, and this study found suboptimal part of KBP and additional AIO might assist in optimization

2. The methods and study design were sound and clear

3. In the discussion, it would be better to include a description of the drawbacks and limitations of this study

Author Response

We would like to thank the reviewer for their comments, which we are sure will strengthen the article.

We have added a paragraph to the discussion regarding the limitations of the study, as suggested.